# Indian-COVID-19 CT Dataset and Analysis of Chest CT Scans of COVID-19 Patients Using Lightweight CNN

**Suba S, Nita Parekh**
Center for Computational Natural Sciences and Bioinformatics
International Institute of Information Technology
Hyderabad, India 500032
`suba.s@research.iiit.ac.in, nita@iiit.ac.in`

**Ramesh Loganathan**
Software Engineering Research Centre
International Institute of Information Technology
Hyderabad, India 500032
`ramesh.loganathan@iiit.ac.in`

**Vikram Pudi**
Data Sciences and Analytics Center
International Institute of Information Technology
Hyderabad, India 500032
`vikram@iiit.ac.in`

**Chinnababu Sunkavalli**
Grace Cancer Foundation
Hyderabad, India
`chinna@gracecancerfoundation.org`

## Abstract

Indian-COVID-19 CT is the chest Computed Tomography (CT) images from COVID-19 patients from India. It has been collected and curated to aid in the diagnosis of COVID-19 and other chest CT analysis tasks using machine learning algorithms. Currently it consists of 6174 images from 142 patients COVID-19, obtained from a single hospital with same image acquisition clinical settings. The dataset will be regularly updated to include more data and the original 3D volumes of dicoms will also be made available. It does not include normal or any other pneumonia images like other similar repositories. It would provide researchers opportunities to develop generalizable and robust models for COVID-19 detection and for developing models for other lung disease detection tasks. To the best of our knowledge, this is the only dataset available from Indian population making it a valuable addition to other similar repositories. Here we also propose a lightweight Convolutional Neural Network (CNN) model to classify chest CT scans into three classes, viz., Normal, non-Covid Pneumonia and COVID-19. The model has been trained and validated on publicly available dataset COVIDx-CT dataset [1]. Performance of the model is evaluated on both COVIDx-CT and Indian-COVID-19 CT datasets and is observed to be comparable, with accuracy slightly lower on Indian-COVID-19 CT dataset. This is not surprising as it is an external test set not seen by the model during training. The proposed lightweight model for diagnosing COVID-19 is well suited for a clinical setting. However, the model is still a prototype and needs more rigorous testing and re-calibrations before using it for clinical diagnosis. The dataset will be made available at http://aimedhub.iiit.ac.in/datasets/gandhi-hospital-covid-dataset.

Submitted to the 35th Conference on Neural Information Processing Systems (NeurIPS 2021) Track on Datasets and Benchmarks. Do not distribute.

# 1 Introduction

With the COVID-19 pandemic shattering the healthcare systems of even the advanced countries the world looks forward to technology for a quick and reliable diagnostic method. Deep learning models have shown their prowess in many fields, so their failing in diagnosing COVID-19 miserably is unexplainable. This can be mainly attributed to the non-availability of reliable data. Numerous studies have been published since the pandemic was declared officially in March, 2020. A good review by Roberts et al (2021) discusses number of reasons why machine learning approaches have been unreliable in a clinical setting [2]. In this study we attempt to address some of the issues in the diagnosis of COVID-19. An alternate diagnosis tool to RT-PCR (Reverse Transcriptase-Polymerase Chain Reaction) is desirable and using chest radiographs to aid in triaging the patients has shown to fulfil the promise. Though chest X-rays (CXR) is a primary option and cheaper, CT scans have a higher sensitivity in diagnosing COVID-19 compared to CXRs [3]. Though sensitive and quick in diagnosing COVID-19, unwarranted use of CT scans should be avoided, and appropriate precautions taken in order to minimize the radiation burden. The study by Kwee and Kwee [4] suggests the use of low-radiation-dose CT instead of full-radiation-dose CT for evaluating the lungs based on the "as low as reasonably achievable" (ALARA) principle to improve the clinical utility of CT scans. Another limitation to the use of CTs is the cost associated with the infrastructure setup thereby making it non-accessible to under-privileged sections of the society.

Indian-COVID-19 CT data is collected from Gandhi Hospital, Hyderabad, India from the COVID-19 isolated patients during the period April - September, 2020. It currently consists of 6174 images from 142 patients at different stages of the disease. The raw dicom files obtained from Gandhi hospital also included CT scans of other organs such as head and abdomen and were removed. Further, for analysis, the dicom slices from 40 to 300 were chosen as these slices contained broad and clear lung window without any other interfering organs. The chosen slices were then converted to png format, in a similar format as other repositories, e.g., COVIDx-CT. A sample image from the dataset is given in Fig.1 along with normal and pneumonia images from COVIDx dataset. No other image augmentations were applied on the dataset as this may introduce additional noise in the data.

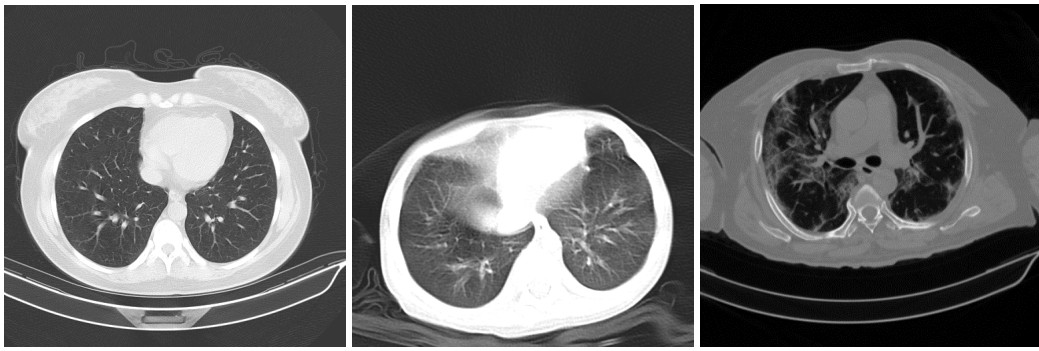

Figure 1: A representative chest CT scan of normal (left), pneumonia (middle) images from COVIDx dataset and COVID-19 image from Indian-COVID-19 CT dataset (right).

There are two major contributions of this work:

1. providing a unique COVID-19 CT scan images of Indian patients, and

2. a lightweight CNN model proposed for the diagnosis of COVID-19. Performance analysis of the proposed model includes analysis on two datasets.

3. performance comparison with deep learning models such as VGG-16, ResNet-50, Inception-v3 and EfficientNetB7 on the proposed dataset.

# 2 Related Works

Chest CT scans are now being extensively used in hospitals as an alternative triaging tool for the diagnosis of COVID-19 as it is sensitive and gives results immediately compared to RT-PCR. Many recent studies have shown that analysis of chest CTs using deep learning methods can reveal even

Table 1: Number of images in the three classes in COVIDx dataset used for training, validation and testing the model. Number of patients are given in brackets.

| Type | Normal | Pneumonia | Covid | Total |
|------|--------|-----------|-------|-------|
| Train | 35996 (321) | 25496 (558) | 82286 (1958) | 143778 (2837) |
| Val | 11842 (126) | 7400 (190) | 6244 (166) | 25486 (482) |
| Test | 12245 (126) | 7395 (125) | 6018 (175) | 25658 (426) |

the most subtle patterns in lung images with comparative or better efficiency than that of expert radiologists. COVIDNet-CT model has gained wide attention in classifying CT scans into Normal, non-Covid Pneumonia and COVID-19 on a hold-out test set with an accuracy of 99.1% [1]. It uses a machine-driven design exploration strategy for building the model with ResNet type backbone that has been pre-trained on ImageNet [5]. The design exploration leverages generative synthesis to identify the network architecture by solving a constrained optimization problem strategy involving spatial, point-wise and depth-wise convolutions. Another study which distinguishes COVID-19 from viral pneumonia uses a pre-trained InceptionNet to convert the image features into a one dimensional vector which is fed as input to a two layered fully connected network [6]. The study uses an external validation dataset to check the performance of the binary classifier. It is shown to achieve an accuracy of 79.3%, specificity 0.83 and sensitivity 0.67 on the external test data. The study by Ardakani et al [7] tested the performance of 10 different CNN architectures in classifying COVID-19 and non-COVID-19 CTs and Resnet-101 was found to have a sensitivity of 100% . The CT images were subjected to annotations by radiologists and the patches of infected areas were extracted and fed to the models. The performance evaluation of the models was done only on a hold-out validation set. Features generated using a CNN along with clinical data such as age, sex, exposure history, symptoms and laboratory tests were integrated in a study to predict COVID-19 [8]. In this study only the CT slices that were identified to have lung infection were used for training the model in classifying positive and negative COVID-19 classes. It achieved a sensitivity of 84.3% and specificity 82.8% on a hold-out test set.

## 3   Dataset Construction

A total of 533 patient data was obtained from Gandhi hospital of which 255 patient data were considered for this study. On initial screening of the 255 samples, 113 patient data were removed as these did not exclusively belong to chest CT, or had missing information like SliceLocation, or came from different CT scanner, and the rest of 142 were subjected to pre-processing. The remaining data of 278 patient samples is under the pre-processing stage and will eventually be added to the Indian-COVID-19 CT dataset. Each CT volume was converted to png format after selecting only slices in the range 40 - 300 as this range was found to consist of the broadest lung window devoid of other internal organs. This heuristic could be applied on all the images as these are obtained from a single CT scanner machine. Every 3rd slice from the chosen range was considered for analysis to reduce the size of the dataset. For a few samples (< 10), however, since sufficient number of slices were not available, every slice in the corresponding range was taken. The images are plain CT scans captured with no contrast and slice thickness of the images are 0.6, 1.5 and 5 mm. The age of the patients is in the range 17 - 79 years with mean age 48 years. The manufacturer's details of the CT machine used is given in the supplementary file, S1. This Indian-COVID-19 CT data is used as an external test set of covid class for evaluating the generalizability of the proposed CNN model. The images in the png format and also the 3D volumes of the data in dicom format will be made available at the dataset link. The details of how to access the dataset and the code for reproducing the results is made available in the supplementary file, S1.

A publicly available benchmark dataset for chest CT classification, COVIDx-CT, has been used in this study. The COVIDx-CT dataset consists of 194922 CT slices from 3745 patients. It has been split into 60-20-20 ratio for training, validation and testing the proposed model, as summarized in Table 1. The number of patients is given in brackets. The respective sources of the data and their publication citations are also mentioned in the supplementary file and consents obtained for the individual sources can be found in the respective publication. No personal identification information or offensive content is contained in the COVIDx data.

## 4 Model architecture

The basic architecture of the proposed model to classify chest CTs into three classes, viz., normal, non-covid pneumonia and COVID-19, is given in Figure 2. It consists of 6 convolutional blocks, with the first block having 16 filters followed by 32, 64, 128, 256 and 512 filters in successive blocks. All kernels are of size 3x3 and a zero padding was used to make the input and output width and height dimensions the same. A 'maxpool' layer was added after first convolution layer and a 'batch normalization' followed by 'max pool' layer added for the remaining five convolutional layers. A dropout layer was added after the fourth, fifth and sixth convolutional layers to avoid overfitting. The convolutional layers were followed by dense layers with 512, 128, 64 and 1 nodes in each layer. Dropout layers were also used after each dense layer. The output layer had a 'softmax' activation function and previous layers of convolution and dense layers used 'Relu' and loss function used was 'categorical cross entropy'. The input image dimensions are 224 x 224 x 3. The hyperparameters of the proposed model such as number of layers, dropout, number of epochs, etc were chosen empirically. Batch size of 8 was chosen based on the capacity of the hardware resources available.

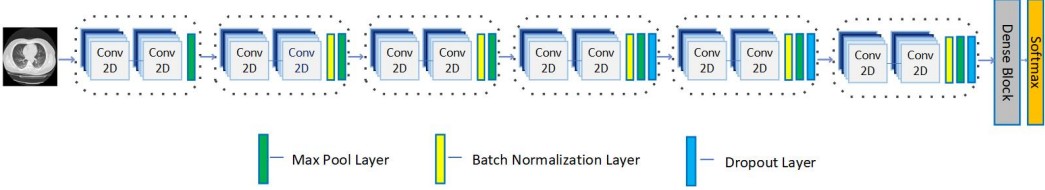

Figure 2: Architecture of the proposed model. The model consists of six convolution blocks marked with dotted rectangles with each block having two convolutional layers. The Max pool, batch normalization and dropout layers are colour coded as shown.

## 5 Implementation

The model was trained on 4 GeForce GTX 1080 Ti GPUs of the internal cluster of IIIT, Hyderabad. and the time taken for training was 36 hours for 18 epochs. The optimizer used was Adam with an initial learning rate set to 5e-6, decay rate of first and second moments were set to the default values of 0.9 and 0.999, respectively. The learning rate was set to reduce by 0.3 if no improvement in validation loss observed for 2 epochs. With the initial learning rate set to the default value of Keras API (= 0.001), the model exhibited very low training and validation accuracy ( 0.46) and did not improve further. So, the cyclic learning rate policy proposed in [9] was implemented and the lower and upper bounds of the optimal learning rate for this system were found to be 5e-3 and 5e-6.

## 6 Results

The proposed CNN model was trained on COVIDx-CT data for 18 epochs and resulted in training accuracy of 0.92, validation accuracy 0.90 and test accuracy 0.92. Other metrics used for evaluating the performance include precision, recall and F1-score, and the results are summarized in Table 2. It may be noted that even for a simple lightweight CNN model proposed here, the precision (0.87) and recall (0.94) values are comparable to that of machine-generated COVIDNet-CT model, precision (0.96) and recall (0.96). The model when evaluated on the external cohort, Indian-COVID-19 CT data gave an accuracy of 0.85, precision 0.82, recall 0.63 and F1-score (0.71). From the confusion matrix (not given) we observe that though majority of COVID-19 cases are being identified correctly by the model, a large number of cases are getting predicted as Normal, resulting in low recall value. This may be due to the fact that the data is from patients from various stages of the disease and in the initial stages, the infection in lungs is not identifiable and hence are predicted as Normal by the model. Another possible reason for low recall is that not all the slices of the CT scan from a patient may exhibit abnormality, and hence predicted by the model as Normal. Performance comparison of the proposed CNN with other state of the art deep learning models such as VGG-16, ResNet-50, Inception-v3 and EfficientNetB7 was carried out on both COVIDx-CT and Indian-COVID-19 CT datasets. The performance of the four DL models for the metrics, Precision, Recall and F1-score are

Table 2: Performance evaluation of CNN and other DL models on COVIDx-CT test data

| CNN | | | | VGG-16 | | | |
|---|---|---|---|---|---|---|---|
| Type | Precision | Recall | F1-score | Type | Precision | Recall | F1-score |
| Covid | 0.87 | 0.94 | 0.90 | Covid | 0.89 | 0.89 | 0.89 |
| Normal | 0.92 | 0.94 | 0.93 | Normal | 0.96 | 0.96 | 0.96 |
| Pneumonia | 0.98 | 0.90 | 0.93 | Pneumonia | 0.94 | 0.94 | 0.94 |
| ReNet-50 | | | | Inception-v3 | | | |
| Type | Precision | Recall | F1-score | Type | Precision | Recall | F1-score |
| Covid | 0.98 | 0.98 | 0.98 | Covid | 0.82 | 0.84 | 0.83 |
| Normal | 0.99 | 0.99 | 0.99 | Normal | 0.96 | 0.95 | 0.95 |
| Pneumonia | 0.99 | 1.00 | 0.99 | Pneumonia | 0.89 | 0.88 | 0.88 |

summarized in Tables 2 and 3 for the two datasets A consistent drop in the performance of all the models on Indian-COVID-19 CT dataset is observed compared to that on COVIDx-CT dataset. This is not surprising as this is an external cohort, not seen by the model during training.In Figure 3, the accuracy, precision and recall of the four models on Indian-CT dataset is depicted (the performance of EfficientNetB7 was very low and not shown). It may be noted that all the three DL models achieved high accuracy by 3 epochs.

## 7 Discussion

This study was carried out with two objectives: to contribute a new dataset to the community that can be used to develop and build better models mainly for the diagnosis/classification of COVID-19 and to compare the performances of the deep learning models on the proposed dataset. The deep learning models, viz., VGG-16, ResNet-50, Inceptio-v3 and EfficientNetB7 along with the proposed lightweight CNN model were trained and tested on the publicly available COVIDx-CT dataset. Performance of these models was also evaluated on an external cohort that is different from the dataset used for training. The objective of this exercise was to indicate the generalizability of the models. Performance metrics used for evaluation are accuracy, precision, recall and F1-score. It is observed that the performance of our lightweight model as well as all the DL models was lower on the proposed dataset compared to the COVIDx-CT dataset used for training. This is not surprising as the data is not seen before by the model. However, it is worth noting that the accuracy of the lightweight CNN (85%) is comparable, in fact marginally better than the three DL models on Indian-COVID-19 CT dataset: ResNet-50 (81%), VGG-16 (83%), Inception-v3 (82%) and EfficientNet (23%). The testing on an external cohort shows the generalizability of these ML models in a real scenario. High recall values of the proposed CNN model on COVIDx-CT dataset for all the three classes in Table 3 indicate fewer false negatives. However, for the Indian-COVID-19 CT test data the recall values of Normal and Pneumonia classes are > 90% but for COVID-19 class slightly lower, which is not surprising as the data is not seen before by the model. The lower recall value and the number of COVID-19 images getting predicted as normal could be because of variation in the severity of the disease between patients and that not all COVID-19 patients may have severe infection in the lungs. This is especially true in the early stages of infection. Apart from the one of its kind Indian data available publicly, the Indian-COVID-19 CT dataset can be useful for other analyses, namely, in training ML algorithms for the detection of lung abnormalities in general, training ML algorithms for detecting COVID-19 disease, as an Indian population-specific external cohort dataset for testing the generalizability of ML algorithms, etc. The dataset can also be used for developing applications for segmentation of lungs and segmentation of the infections at the slice level. As there is scarcity of data from the Indian population the dataset can also help in generating new datasets using generative models. Slice level classification models based on the presence or absence of infection in the slices is yet another application for which the data can be used for.

In this study we have attempted to follow the recommendations proposed M Roberts et al [2] in constructing the dataset, training the model and also in evaluating the performance of the model to reduce bias at every stage of the analysis from data collection to the final outcome. For training, only CTs that are RT-PCR or radiologist confirmed true COVID-19, have been considered and the

Table 3: Performance evaluation of DL models on Indian-COVID-19 CT test data. In the test data Normal and COVID-19 images are taken from COVIDx-CT. The confidence interval is given in brackets for precision and recall.

| CNN | | | |
|---|---|---|---|
| Type | Precision | Recall | F1-score |
| Covid | 0.82 | 0.63 | 0.71 |
| (CI%) | (81.3, 83.5) | (61.5, 63.9) | |
| Normal | 0.80 | 0.94 | 0.86 |
| (CI%) | (78.9, 80.2) | (93.2, 94.0) | |
| Pneumonia | 0.99 | 0.90 | 0.94 |
| (CI%) | (98.3, 98.9) | (88.9, 90.3) | |

| VGG-16 | | | |
|---|---|---|---|
| Type | Precision | Recall | F1-score |
| Covid | 0.81 | 0.44 | 0.57 |
| (CI%) | (79.5, 82.2) | (42.3, 44.7) | |
| Normal | 0.83 | 0.96 | 0.89 |
| (CI%) | (82.1, 83.3) | (96.0, 96.7) | |
| Pneumonia | 0.84 | 0.94 | 0.89 |
| (CI%) | (83.2, 84.8) | (93.0, 94.1) | |

| ReNet-50 | | | |
|---|---|---|---|
| Type | Precision | Recall | F1-score |
| Covid | 0.92 | 0.22 | 0.36 |
| (CI%) | (90.5, 93.3) | (21.0, 23.0) | |
| Normal | 0.91 | 0.99 | 0.95 |
| (CI%) | (90.1, 91.1) | (98.5, 98.9) | |
| Pneumonia | 0.67 | 1.00 | 0.80 |
| (CI%) | (66.0, 67.8) | (99.3, 99.6) | |

| Inception-v3 | | | |
|---|---|---|---|
| Type | Precision | Recall | F1-score |
| Covid | 0.73 | 0.48 | 0.58 |
| (CI%) | (71.8, 74.5) | (46.7, 49.2) | |
| Normal | 0.85 | 0.95 | 0.90 |
| (CI%) | (84.7, 85.9) | (94.7, 95.5) | |
| Pneumonia | 0.80 | 0.88 | 0.84 |
| (CI%) | (79.2, 80.9) | (87.1, 88.6) | |

external test dataset, Indian-COVID-19 CT dataset, has been collected from the hospital in Hyderabad through assigned, reliable sources and confirmed to be of only COVID-19 positive patients. The demographics of the training, validation and test datasets are compared, and the range of patients age, mean age of the patients, etc. are found to be comparable across the two datasets. To address the issue of bias, if any, in the outcome, the model is tested on a completely different dataset from the one used for training. The test performance indicates that the proposed model is generalizing well and there is no data dependent bias affecting the outcome of the study. In fact, the performance on the lightweight CNN model on the external dataset is marginally better compared to the deep learning models, indicating its reliability in the clinical setting as an alternative diagnostic tool for triaging the patients. However, there is a bias introduced in the training phase due to higher number of COVID-19 images ( 82k) compared to normal ( 35k) and pneumonia classes ( 25k). Yet another limitation is that the external cohort now has only COVID-19 data.

Indian-COVID-19 CT is the only dataset available from India. The characteristic feature of this dataset is that all the images are from the same hospital, from the same place (i.e., Hyderabad) and generated under identical settings (same scanner). On the other hand, the largest publicly available dataset, COVIDx-CT is built from multiple sources from over a dozen countries. The fact that it is

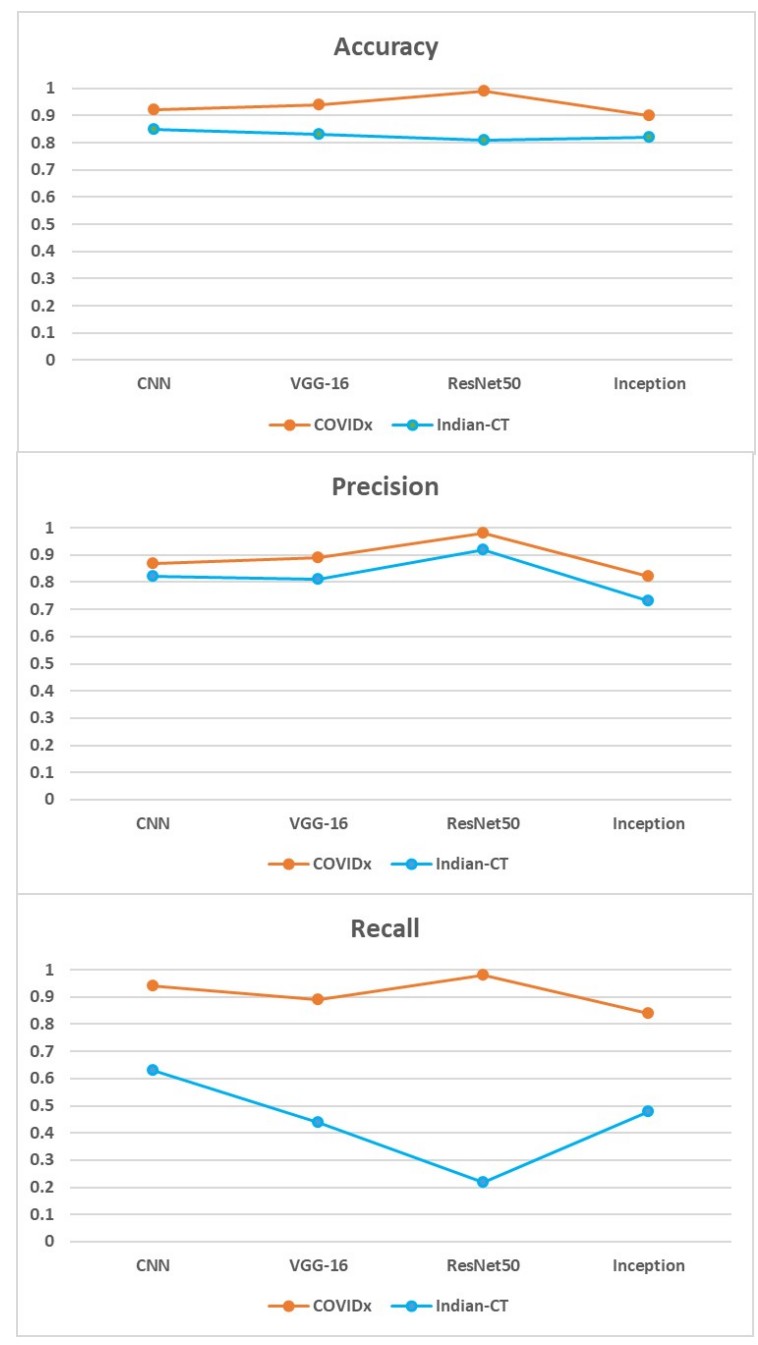

Figure 3: Accuracy, precision and recall plots for CNN and other DL models.

obtained from India alone makes it suitable for studies from Indian population and also the chances
of various other factors confounding data for a research study such as different machine settings,
different living conditions, etc. are absent. This makes it a very useful dataset for evaluating the
performance of algorithms.

However, this dataset has some inherent limitations too, the fact that these images are only from a
small region from a vast country like India and all the images are obtained from a single CT scan
machine. This will bring in some associated biases as well. Since there is scarcity of publicly available
medical image data in general and are rarely from a country like India, this dataset is valuable for

the research and machine learning communities in understanding the disease and developing more generalizable models.

## Acknowledgments and Disclosure of Funding

This work had been funded partly by the RAKSHAK (Remedial Action, Knowledge Skimming and Holistic Analysis of COVID-19) project of Department of Science and Technology (DST), India. We would like to thank the team from Gandhi Hospital, Hyderabad and the DST for the support and the discussions provided.

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
