# OpenReview forum: "Indian-COVID-19 CT Dataset and Analysis of Chest CT Scans of COVID-19 Patients Using Lightweight CNN "
_NeurIPS.cc/2021/Track/Datasets_and_Benchmarks/Round1 — Submitted to NeurIPS 2021 Datasets and Benchmarks Track (Round 1)_

### Official Review · Reviewer_Ucsj · 2021-06-25
**Indian COVID CT dataset**

**Rating:** 1
**Confidence:** 4
**Clarity:** This is a well written paper.

**Strengths:**

Well written paper.  From a very strict point of view, the proposed dataset is novel in the sense that it is the only one from India.

**Weaknesses:**

The proposed dataset is one to two orders of magnitude smaller than the COVIDx CT-2A dataset (https://arxiv.org/pdf/2101.07433.pdf) which comprises 194,922 CT images  of 3,745 patients.  This contribution is thus not very significant and of low interest for the community.  Same with the proposed model which is *very* naive.  At least, they should have compared with other light-weight models like EfficientNet or SqueezeNet.  Also, the study of the learning rate in Table 6 has very little interest.  LR is just an hyper-parameter that people get to tweak.  No need in a paper like this one to report results of their hyperparameter search.


**Additional Feedback:**

This paper is way below the NeurIPS standards.  Clear reject.

**Correctness:**

Authors claim two things:

* *their dataset is unique*.  While this is true in the sense that this dataset is the only one from India, others like COVIDx-CT (which happens to be much larger) have been released before.  In fact, the proposed dataset is so small that the authors did not trained on it.
* *new light-weight CNN model*.  This model is trivial and was not compared thoroughly against other SOTA models.

**Documentation:**

The dataset is not available as of today.  The authors refer to an IIT website which does not seams to have neither a submission system nor a leaderboard.  Not clear who is behind that website and how long it will be hosted.  There is also no mention to any licensing.  Does that mean anybody can use it for free?

**Ethics:**

Hard to tell.  Besides the fact that the images have been put in png format, no ethical topic has been mentioned.  Was this project evaluated by a local ethical committee?  Not clear.

**Relation To Prior Work:**

The paper does not mention how the proposed dataset differs from previous ones (besides the fact that it comes from India).

**Summary And Contributions:**

This paper presents a new COVID19 CT dataset acquired in India (Ghandi Hospital).  This dataset comprises 147 patients and 6174 images classified in 3 classes : normal, pneumonia and COVID19.

The authors also introduce a light-weight (yet very simple) CNN which they trained on another dataset called COVIDx-CT.  In this context, the indian dataset is used as a testset.  Accuracies on the latter is lower than on the former, probably due to the inter-class imbalance of both datasets.

---

> ### Author Response · Authors · 2021-07-15
> **Indian-COVID-19 CT Dataset is now made available and more rigorous empirical analysis carried out**
>
> **Summary And Contributions**
>
> ->We would like to make a clarification  - the Indian-COVID-19 CT dataset consists of 6174 images from 142 COVID-19 patients. No normal or pneumonia patients are included in this set. To remove any confusion, we rename it as Indian-COVID-19 CT dataset in the revised manuscript.
>
> ->It is explicitly mentioned in the manuscript in the first line, 2nd para of Introduction.
>
> ->Training, validation, and testing of the CNN model was carried out on COVIDx-CT dataset and Indian data was used as an external cohort for only Covid-19 class. Test accuracy on COVIDx-CT dataset was 0.92 and slightly lower on Indian-COVID-17 CT dataset, \~0.85. The COVID-19 image data in the Indian-COVID-19 CT dataset is from a different population than the one used for training and new to the model, hence the drop in accuracy. Further, since the Indian-CT data consists of patients from different stages of the disease,  a large number of COVID-19 images (~2287) were predicted as Normal in the Indian-COVID-19 CT data. This is probably because of no significant abnormalities in the CT during the early stages of the disease, while only 295 COVID-19 images were predicted as normal in case of COVIDx-CT. This is another possible reason for the drop in accuracy on Indian-CT data.
>
> ->The revised lines given in lines 137 - 143 in the revised manuscript in the Results section, page no 4 and reproduced below.
>
> **Weakness**
>
> * The number of COVID-19 images in COVIDx was 94,548 from 2299 patients. The Indian-CT dataset is the only one of its kind data from one local region of India (i.e., Hyderabad) and will be available publicly. Availability of medical imaging data is rare due to various reasons and hence COVID-19 CT-image data is available from only a few countries and with this dataset, India is one of the country hosting COVID-19 image data.
>
> * Our objective of proposing a simple CNN model was to show that it is able to achieve comparable performance with other Deep learning models and better suitable in a clinical setting. The proposed Indian-COVID-19 CT dataset was used here to show that it can be used for testing the generalizability of Machine Learning models.
>
> * The analysis of the proposed CNN has been carried out with other models including, EfficientNet, Inception-v3, VGG-16 and ResNet-50 and is now added in the revised manuscript. Line No 166 in Discussion Section, page no 5 and Tables 2 & 3.
>
> * We agree with the reviewer’s feedback regarding learning rate tuning. It is now removed from the revised manuscript.
>
> **Correctness**
>
> * COVIDx-CT dataset is a large cohort built from multiple data sources from China, Iran, Italy, Turkey, Ukraine, Belgium, Australia, Afghanistan, Scotland, Lebanon, England, Algeria, Peru, Azerbaijan and has images from 2299 COVID-19 patients. Indian-COVID-19 CT is the only COVID-19 dataset to the best of knowledge that is being made publicly available from India and has currently images from 142 COVID-19 patients. It will soon be updated after pre-processing of 278 patient data. The dataset was used only for testing as it contains data from only COVID-19 patients and has no data from normal and pneumonia patients. Hence it was not considered for training. Also, we were interested in showing the generalizability of the proposed CNN model on a dataset not seen by the model during the training phase.
>
> * The model was proposed as a suitable model in a clinical setting for such healthcare applications and as a tool to test the performance of Machine Learning models on the proposed dataset. Performance comparison of the proposed CNN model with other state-of-the-art models has been carried out and the results are summarized now in the Tables 2 & 3 on page no 5 & 6.
>
> **Relation To Prior Work:**
>
> * We thank the reviewer for the valuable feedback. We have revised the manuscript incorporating the changes suggested. These are updated in the revised manuscript.
>
> * Indian-CT is the only dataset available from India. The characteristic feature of this dataset is that all the images are from the same hospital, from the same place (i.e., Hyderabad) and from identical clinical settings (same scanner). On the other hand, the largest publicly available dataset, COVIDx-CT is built from multiple sources from over a dozen countries. As correctly pointed out, the fact that it is obtained from India alone makes it suitable for studies from Indian population and also the chances of various other factors confounding data for a research study such as different machine settings, different living conditions, etc. are absent. This makes it a very useful dataset for evaluating the performance of algorithms.
>
> **Documentation**
>
> * Details regarding access to the data is updated in Datasheets in Supplementary file.
>
> **Ethics**
>
> * It is mentioned in line number 149 in the datasheet in supplementary file regarding the Institutional Review Board approvals.

---

> > ### Comment · Reviewer_Ucsj · 2021-07-19
> > **final review**
> >
> > The answer confirm my original review.  As such, I keep my initial review. i.e.
> >
> > 1- Trivial or wrong
> >
> > I would recommend the authors to submit their paper to a less selective and more medical-oriented venue.

---

### Official Review · Reviewer_NwAb · 2021-07-03
**Important dataset but paper needs more work**

**Rating:** 3
**Confidence:** 4

**Strengths:**

The strengths of this paper lie in the rarity of the data that has been collected. Most healthcare datasets which are publicly released are from North America or the UK. Datasets from other regions / countries are rarely seen. It is important for the ML4H community to continue methodological development and validation on datasets from a variety of regions around the world. This dataset would be one of the few medical imaging datasets from India that is publicly available. Additionally, the study is thorough in its analysis of different metrics and issues such as generalizability and bias.

**Weaknesses:**

The paper is not clear and needs some more work on the writing. Additionally, as one of the goals of this paper appears to be to serve as a benchmark for additional methods development and validation their empirical analysis needs to be more rigorous. Currently, only point estimates of performance are provided without confidence intervals. Additional models / model architectures should be tried as well to serve as a starting point for future developments.


**Additional Feedback:**

- The study describes the failure of deep learning models to diagnose COVID-19 as unexplainable but I would challenge that this is expected. This is a disease for which he had no data on until recently. This is discussed in the following sentence but I think the wording could be better.
- The authors describe CT as a desirable alternative to RT-PCR but a quick sentence / discussion on the risks vs benefits would be good. For example, CT comes with a large amount of radiation.
- This paper is important and I would like to see it published in a future venue. There is a lack of datasets in medicine from regions such as India and this is stalling the benefits of machine learning in healthcare globally.
- More rigour in the empirical analyses and improved writing need to be seen.


**Clarity:**

The writing of the paper needs a lot more work. There are a number of grammar, formatting, and spelling issues which need to be improved. I think with improvements to the writing this paper would be much stronger. Currently, the lack of clarity takes away from the paper unfortunately.


**Correctness:**

To the best of my knowledge, everything presented is correct.


**Documentation:**

The dataset curation was well documented. Additionally, the code used to train the benchmark models is well documented and the infrastructure used to train the model was done well.


**Ethics:**

There are no ethical concerns that warrrant further discussion or review from my perspective. Based on the paper it seems like the proper precautions were taken with respect to privacy.


**Relation To Prior Work:**

The study attributes prior work well and also makes sure to test their method on datasets curated from prior work such as the COVID-CTx dataset. I appreciate the care taken in ensuring the study builds off of prior work well.

**Summary And Contributions:**

This study constructed a new medical imaging dataset which focuses on the diagnosis of COVID-19 from CTs. The dataset was collected from a hospital in India from 147 patients with different disease severity. The study also provides performance results from training a CNN model on this data described as the Indian-CT dataset and an external test set called the COVIDx-CT dataset. The data is publicly released with a corresponding Github repo for the models trained.

---

> ### Author Response · Authors · 2021-07-15
> **Paper has been rewritten and more empirical analysis reports has been added**
>
> **Weakness**
>
> ->Performance comparison of the proposed CNN model with other machine learning models such as VGG-16, ResNet-50, Inception-v3 and EfficientNet is now added in the revised manuscript. The results are summarized in the Tables 2 & 3 in manuscript along with confidence intervals
>
> **Clarity**
>
> ->In the revised manuscript care has been to remove any grammar and spelling issues.
>
> **Additional Feedback**
>
> * We agree with the reviewer and have specified the same in the following line no: 169 on page no 5, reproduced below:
>
> ->"For the Indian-COVID-19 CT data, the recall values of Normal and Pneumonia classes are > 90% but for COVID-19 class slightly lower, which is not surprising as the data is not seen before by the model.”
>
> * The discussion on risks of using CT has been added in the revised manuscript in line no 35 – 41 on page no 2 and is reproduced below.
>
> ->"Though sensitive and quick in diagnosing COVID-19, unwarranted use of CT scans should be avoided, and appropriate precautions taken in order to minimize the radiation burden. The study by Kwee and Kwee [4] suggests the use of low-radiation-dose CT instead of full-radiation-dose CT for evaluating the lungs based on the "as low as reasonably achievable" (ALARA) principle to improve the clinical utility of CT scans. Another limitation to the use of CTs is the cost associated with the infrastructure setup thereby making it non-accessible to under-privileged sections of the society."
>
> * We are thankful for the valuable feedback provided by the reviewers to help us improve the manuscript. We hope now it meets the standards of NeurIPS.
>
> * More empirical analysis has been carried out using other state of the art machine learning models and the results have been updated in the revised manuscript and reproduced below.
>
> ->"Performance comparison of the proposed CNN with other state of the art deep learning models such as VGG-16, ResNet-50, Inception-v3 and EfficientNetB7 was carried out on both COVIDx-CT and Indian-COVID-19 CT datasets. It may be noted that all the three DL models achieved high accuracy by 3 epochs. The performance of the four DL models for the metrics, Precision, Recall and F1-score are summarized in Tables 2 and 3 for the two datasets. A consistent drop in the performance of all the models on Indian-COVID-19 CT dataset is observed compared to that on COVIDx-CT dataset."

---

### Official Review · Reviewer_VmLV · 2021-07-05
**A new Chest-CT dataset containing COVID-19 patients and a CNN model for the diagnosis of COVID-19**

**Rating:** 4
**Confidence:** 5

**Strengths:**

-	Relevant dataset to augment the existing set of COVID-19 CT data available.
-	A good number of CT patients with COVID-19 (6174 images from 147 patients) are available in the dataset.


**Weaknesses:**

- Lack of clarity on explaining some parts of the dataset construction. Please see below for further details.
- The inherent bias that comes with picking up a cohort specific to a particular population especially in such medical data is not addressed.
- The structure of the paper seems to focus on the proposed CNN model and the authors have dedicated more explanations for that rather than on the dataset.
- The only mentioned use case of the dataset is shown as a proposed holdout dataset and has not explicitly addressed other possible use cases for this cohort.
- No mention of the maintenance of the dataset.
- Missing information on how long would the dataset be available.


**Additional Feedback:**

-	More emphasis on the proposed CNN model which in itself is not sufficiently novel. Readers would have appreciated it more if the authors elaborated on the preparation of the dataset and their vision of how the dataset can be used further.
-	It is not clear why the authors did not consider using the Indian-CT COVID data as part of the training system as well. Since the authors did not compare their proposed model with other state-of-the-art existing networks, it is unclear why they did not attempt to use some part of the Indian-CT dataset in the training. I think adding a comment on this would help readers appreciate the reasoning behind this choice.
-	Conversion to PNG. Although the authors followed the COVIDx-CT dataset, in the example image on Fig.1 it is clear that the author's dataset has been thresholded differently given the brightness. And because the pixel information in CTs is related to Houndsfield Units I would like the authors to comment on the dynamic range which they chose to threshold.


**Clarity:**

Unsatisfactory. I have a few comments on the organization and claims made in the paper:

- Lines 48-49:  *Chest CT scans are now being extensively used in hospitals as primary tool for diagnosis of COVID-19 being more sensitive and in giving immediate results compared to RT-PCR* - Disagree on the primary tool claim. If you see Kwee et. al, [1] and the ACR [2] it is still not recommended that CT be a stand-alone tool but only as an evidence collection tool for cases with a positive RT-PCR test.

- Lines 36-37 & 76-77:  Authors first claim Dicom slices from 40 to 300 were chosen and later on mention that slices in the range of 40-200 were chosen. This discrepancy needs to be addressed and made consistent. Isn't this heuristic only applicable to the hospital's CT scanner settings? If so I would recommend adding a line mentioning that this heuristic only works for that particular scanner setting.

- Lines 74-75: *The remaining data is under the pre-processing stage and will eventually be added to the Indian-CT dataset.* – If so how were the proposed deep learning models run on the paper. Please clarify what pre-processing is still remaining and how the results reported have been reported. Also, include what other pre-processing is being suggested besides png.

- Line 34,35,74: *On initial screening 108 patient data were removed as these did not exclusively belong to chest CT*.
In the introduction, the authors claimed the following: *The raw dicom files obtained from Gandhi hospital also included CT scans of other organs such as head and abdomen, and were removed.* It's not clear how this 108 was different from the pre-processed 147 in which some of the body parts were heuristically removed for analysis. Providing more clarity on these differences will help readers appreciate it better.

- Line 77-79: *Every 3rd slice from the chosen range was considered for analysis to reduce the size of the dataset. However, for some samples sufficient slices were not available and in such cases every slice in the corresponding range was taken.* – How many such datasets versus every 3rd slice were chosen dataset. Giving statistics on this heuristic and also reporting on the average number of slices chosen from every patient CT will be useful for the dataset users.

- Lines 144-147: The central claim of the discussion on the overall paper is the proposed lightweight CNN model. However, the purpose of Dataset track submission is not on the proposed novelty of a CNN. But on the description of new datasets for machine learning and expanding on its preparation, acquisition, and other details. An example use case is certainly encouraged but on reading the premise of the paper I feel it's the other way around and that the paper is more focused on their proposed network with the new Indian-CT dataset used for holdout experiments.

- This track requires suggesting a few use cases for this dataset which is not addressed by the authors.

References:

[1] Kwee, Thomas C., and Robert M. Kwee. "Chest CT in COVID-19: what the radiologist needs to know." RadioGraphics 40.7 (2020): 1848-1865.

[2] https://www.acr.org/Advocacy-and-Economics/ACR-Position-Statements/Recommendations-for-Chest-Radiography-and-CT-for-Suspected-COVID19-Infection

**Correctness:**

The claims on a novel dataset of COVID-19 CT scans on the Indian population are correct. The claims of novelty on their proposed few-layered CNN and its benefits are questionable.

**Documentation:**

-	data collection and organization – The collection parameters and the dataset pre-processing for sampling the png are clear. However, they mention that it would be added when the pre-processing has been completed but it's not clear how the machine learning experiments were run if the dataset was still not processed.
-	availability and maintenance – Authors provide a link for the availability of the dataset but do not mention anything about its maintenance or how long they plan to keep providing the dataset. They do mention more data is being collected and will be added to the same repository.
-	ethical and responsible use – The authors have IRB approval on this dataset. The authors don’t mention what responsible use practices should be for this dataset. However given that this is an anonymized dataset which they have curated, general best deep learning practices on the dataset should be sufficient.


**Ethics:**

No particular ethical concerns. This study had IRB approval and the dataset was anonymized.

**Relation To Prior Work:**

Authors use a previous COVID-CTx dataset and use the same for the training and validation of their proposed network. They do survey a few of the earlier COVID-19 related machine learning networks but do not compare with any of them and mention it as one of the limitations of this study.

**Summary And Contributions:**


The authors propose a new dataset of Chest-CT called Indian-CT consisting of 6174 images from 147 patients including patients with COVID-19 from the cohort consisting of Indian patients. These patients are chosen from a subset of patients from Gandhi Hospital, Hyderabad, India. They demonstrate using this dataset and an external holdout dataset on the proposed COVID-19 classification model trained on a larger COVIDx-CT dataset. Experiments show comparable performance on the validation set and the holdout dataset for the proposed architecture.

---

> ### Author Response · Authors · 2021-07-15
> **Manuscript has been revised and restructured to give more importance to dataset and more use cases added. More analysis has also been done.**
>
> **Weakness**
> * Inherent bias
> ->Now it has been included in the revised manuscript. Line No 207 on page no 7
> * Structure of paper:
> ->We have now restructured the paper to address this point. The Abstract, Dataset construction, and portions of Results and Discussion have been re-written to give importance to the data rather than the model.
> * Usecases:
> ->More use cases mentioned in Line No 174 - 182
> * Maintenance:
> ->Details added in Datasheet in supplementary file
> * Availability:
> ->Will be available for long time (5 years). Any change in the dataset will be informed to collaborators via email as the use is restricted to only collaborators.
>
> **Correctness**
> * Claims of benefits of proposed CNN:
> ->Performance comparison of the proposed CNN with other state-of-the-art methods is now provided. Line No 165, Table 3.
>
> **Clarity**
> * Lines 48-49:
> ->Modified as "Chest CT scans are now being extensively used in hospitals as an alternative triaging tool for diagnosis of COVID-19  as it is sensitive and gives results immediately compared to RT-PCR."
> * Lines 36-37 & 76-77:
> ->Thanks for pointing out the typo; the range of selected slices is corrected across the whole manuscript and the line added
> * Lines 74-75:
> ->Modified in lines 82-98
> * Lines 34,35,74:
> ->Modified in lines 82-87
> * Lines 77-79:
> ->For a small number of patient samples (< 10), the number of slices were fewer and so all the slices were considered for analysis. For the remaining patient samples every 3rd slice was taken. Modified in line 91
> * Lines 144-147:
> ->Manuscript has been revised accordingly
> * Usecases:
> ->Added in lines 178-182 as follows: Apart from the one of its kind Indian data available publicly, the Indian-COVID-19 CT dataset can be useful for other analyses, namely, in training ML algorithms for the detection of lung abnormalities in general, training ML algorithms for detecting COVID-19 disease, as an Indian population-specific external cohort dataset for testing the generalizability of ML algorithms, etc. The dataset can also be used for developing applications for segmentation of lungs and segmentation of the infections at the slice level. As there is scarcity of data from the Indian population the dataset can also help in generating new datasets using generative models. Slice level classification models based on the presence or absence of infection in the slices is yet another application for which the data can be used for.
>
> The datasheets for the dataset is prepared and added to the supplementary file with all the details of data acquisition, preparation and other details. We are thankful for the valuable feedback on the focus of the paper. We have rewritten the paper based on the feedback adding more details about the proposed dataset
>
> **Relation to prior work**
>
> ->In the revised manuscript the proposed CNN is compared with other state-of-the-art deep learning models such as VGG-16, ResNet-50, InceptionNet and EfficientNet and the results are summarized in the Tables 2 & 3.
>
> ->Here we show that the performance of the lightweight model  and the DL models considered, all show a drop in accuracy on the proposed dataset.
>
> ->None of the other studies have published their model for a comparison and if they have, only pre-trained model weights have been published by a few studies. For this reason, we have done only the comparison of their published results with ours. Lines 63-80 and 135
>
> **Documentation**
>
> data collection and organization, availability and maintenance, ethical and responsible use:
>
> ->Added in datasheet.
>
> **Additional Feedback**
>
> ->As mentioned above, the manuscript has been re-written so that the data acquisition and preparation part of the study is given more importance. More use cases on how the dataset can be used have also been added.
>
> ->Indian-CT dataset has only COVID-19 samples, no normal or pneumonia samples. Hence this data was not used for training. Moreover, we wanted to show how the deep learning models generalize on a new dataset. We have now made these points clear in the revised manuscript.
>
> ->We have not selected a particular range of Houndsfield Units (HU) but just converted the voxels in the slices to HU and converted them into the pixels. The rescale intercept and rescale scope values were set at the time of acquisition of the slices, we have just read the values and converted to HU.

---

### Decision · Program_Chairs · 2021-07-26

**Decision:**

Reject

**Comment:**

The reviews feel that the dataset augments the existing set of COVID-19 CT data, but the contributions are not sufficient for acceptance. In particular, the paper focuses on the model rather than on the dataset. The clarity also needs to be improved.